# Febrile Neutropenia Duration Is Associated with the Severity of Gut Microbiota Dysbiosis in Pediatric Allogeneic Hematopoietic Stem Cell Transplantation Recipients

**DOI:** 10.3390/cancers14081932

**Published:** 2022-04-12

**Authors:** Riccardo Masetti, Federica D’Amico, Daniele Zama, Davide Leardini, Edoardo Muratore, Marek Ussowicz, Jowita Fraczkiewicz, Simone Cesaro, Giulia Caddeo, Vincenza Pezzella, Tamara Belotti, Francesca Gottardi, Piero Tartari, Patrizia Brigidi, Silvia Turroni, Arcangelo Prete

**Affiliations:** 1Pediatric Oncology and Hematology “Lalla Seràgnoli”, IRCCS Azienda Ospedaliero-Universitaria di Bologna, 40138 Bologna, Italy; riccardo.masetti5@unibo.it (R.M.); edoardo.muratore@studio.unibo.it (E.M.); tamara.belotti@aosp.bo.it (T.B.); francescag.ageop@aosp.bo.it (F.G.); piero.tartari@aosp.bo.it (P.T.); arcangelo.prete@aosp.bo.it (A.P.); 2Department of Medical and Surgical Sciences (DIMEC), University of Bologna, 40138 Bologna, Italy; federica.damico8@unibo.it (F.D.); daniele.zama2@unibo.it (D.Z.); patrizia.brigidi@unibo.it (P.B.); 3Department of Pharmacy and Biotechnology (FABIT), University of Bologna, 40126 Bologna, Italy; silvia.turroni@unibo.it; 4Pediatric Emergency Unit, IRCCS Azienda Ospedaliero-Universitaria di Bologna, 40138 Bologna, Italy; 5Department and Clinic of Pediatric Oncology, Hematology and Bone Marrow Transplantation, Wroclaw Medical University, 50-556 Wrocław, Poland; ussowicz@o2.pl (M.U.); jowita.fraczkiewicz@umed.wroc.pl (J.F.); 6Department of Pediatric Hematology Oncology, Azienda Ospedaliera Universitaria Integrata, 37126 Verona, Italy; simone.cesaro@aovr.veneto.it (S.C.); giulia.caddeo@aovr.veneto.it (G.C.); vincenza.pezzella@aovr.veneto.it (V.P.)

**Keywords:** febrile neutropenia, gut microbiome, hematopoietic stem cell transplantation, *Akkermasia*

## Abstract

**Simple Summary:**

Febrile neutropenia is a common complication in pediatric patients undergoing allogeneic hematopoietic stem cell transplantation. Its genesis is often attributed to infections; however, a specific cause frequently cannot be defined. We hypothesize that the composition of the intestinal flora may contribute to the genesis of the neutropenic fever. We analyzed the microbial composition of stool samples from pediatric patients from three European centers and assessed the relationship with the duration of the fever during neutropenia. We found that a more stable composition of the microbiota during the transplantation course is associated with a shorter duration of fever. Moreover, patients with a higher duration of fever presented higher levels of *Collinsella*, *Megasphaera*, *Prevotella*, *Roseburia*, *Eggerthella* and *Akkermansia* in the stool.

**Abstract:**

Febrile neutropenia (FN) is a common complication in pediatric patients receiving allogeneic hematopoietic stem cell transplantation (HSCT). Frequently, a precise cause cannot be identified, and many factors can contribute to its genesis. Gut microbiota (GM) has been recently linked to many transplant-related complications, and may also play a role in the pathogenesis of FN. Here, we conducted a longitudinal study in pediatric patients receiving HSCT from three centers in Europe profiling their GM during the transplant course, particularly at FN onset. We found that a more stable GM configuration over time is associated with a shorter duration of fever. Moreover, patients with longer lasting fever exhibited higher pre-HSCT levels of *Collinsella*, *Megasphaera*, *Prevotella* and *Roseburia* and increased proportions of *Eggerthella* and *Akkermansia* at the engraftment. These results suggest a possible association of the GM with the genesis and course of FN. Data seem consistent with previous reports on the relationship of a so-called “healthy” GM and the reduction of transplant complications. To our knowledge, this is the first report in the pediatric HSCT setting. Future studies are warranted to define the underling biological mechanisms and possible clinical implications.

## 1. Introduction

Fever during neutropenia, (hereinafter termed as febrile neutropenia (FN)), is almost universally present in pediatric allogeneic hematopoietic stem cell transplantation (HSCT) recipients [1]. Bloodstream infections (BSI) represent the most frequent cause of fever, complicated by high mortality, particularly in cases related to Gram-negative bacteria [2,3]. However, only in half of the patients can an actual infectious etiology be identified [2]. Indeed, fever can also be attributed to other factors, including viral and fungal infections, graft-versus-host disease (GvHD) and cytokine release [4].

Growing evidence is pointing to a pivotal role of the microbial community inhabiting the gastrointestinal tract (i.e., the gut microbiota, GM) during anticancer therapies, and particularly during allogenic HSCT (allo-HCST) [5]. The disruption of the symbiosis between the host and the GM is well documented in pediatric patients receiving allo-HSCT [6,7], mainly due to many factors acting early in the neutropenic period, such as antibiotic administration, nutritional deficiencies, immunosuppression and chemotherapy-related toxicity [8,9,10,11,12]. In recent years, the relationship between GM and allo-HSCT outcomes has been extensively explored, with a particular focus on overall survival [13], relapse [14], GvHD [15], veno-occlusive disease [16] and BSI [6,17,18]. However, whether intestinal commensal bacteria could have a role in the genesis and course of FN has not yet been addressed. In transplanted adults, GM profiles have been observed to diverge between patients who developed FN and those who did not [19]. In particular, patients with FN showed enrichment in *Mogibacterium*, *Bacteroides fragilis* and *Parabacteroides distasonis*, while they showed a depletion in *Prevotella*, *Ruminococcus*, *Dorea*, *Blautia* and *Collinsella*. Recent evidence has also suggested that *Akkermansia* expansion in the gut of adult patients with acute leukemia could be a predictive signature for FN development [20]. However, evidence in the pediatric setting is still lacking to date. Improving HSCT outcomes could represent a crucial issue in improving outcomes of high-risk cancer patients since often it represents the best treatment option. Thus, the aim of this study was to characterize the GM dynamics in pediatric patients undergoing HSCT, with a focus on the neutropenic phase and the relationship between GM configurations and FN characteristics.

## 2. Materials and Methods

### 2.1. Study Population and Stool Sampling

This prospective, multi-center study enrolled pediatric patients undergoing allo-HSCT in three pediatric centers in Europe (Bologna, Italy; Verona, Italy; Wroclaw, Poland) between January 2019 and December 2020. Inclusion criteria were age ≤ 18 years and informed consent provided. Exclusion criteria included previous fecal microbiota transplantation. Enrolled patients could be candidates for HSCT for any underlying condition, with any HSCT characteristics. The sampling protocol was approved by the Ethics Committee of the CE-AVEC of Emilia-Romagna, Italy (ref. number 19/2013/U/Tess). The study was conducted in accordance with the Declaration of Helsinki. Stool samples were collected at baseline (before transplant; hereafter referred to as PRE); at the onset of neutropenia (day −2/+2; P2); at the onset of fever (day +4/+5; P5); at engraftment (absolute neutrophil count (ANC) ≥ 500 × 10^6^/L for at least 3 days; TAKE) and after engraftment (day +20/+30; P20). Stool samples were collected in sterile tubes and stored at −80 °C and shipped in dry ice to the Department of Pharmacy and Biotechnology (FABIT), University of Bologna, Italy where the analysis was performed. Neutropenia was defined as ANC < 500 × 10^6^/L and fever was defined as a body temperature ≥ 38 °C proven by two different determinations. Episodes of BSI were defined as positive cultures from a blood sample from both the central venous catheter and peripheral veins.

### 2.2. Microbial DNA Extraction and 16S rRNA Gene Library Preparation

Microbial DNA was extracted from 0.25 g of fecal sample using the repeated bead-beating protocol [21], with only a few modifications [22]. In short, stool samples were resuspended in 1 mL of lysis buffer in presence of four 3-mm glass beads and 0.5 g of 0.1-mm zirconia beads (BioSpec Products, Bartlesville, OK, USA), and bead-beaten in a FastPrep homogenizer (MP Biomedicals, Irvine, CA, USA) at 5.5 movements/s for 1 min three times. After 15-min incubation at 95 °C, supernatants were separated by centrifugation at 13,000 rpm for 5 min and incubated first with 260 µL of 10 M ammonium acetate and then with one volume of isopropanol for 30 min. The nucleic acid pellets were washed with 70% ethanol and resuspended in 100 µL of TE buffer. RNA was removed by 15-min incubation at 37 °C with 2 µL of DNase-free RNase (10 mg/mL). For the subsequent DNA purification steps, the DNeasy Blood and Tissue Kit (QIAGEN, Hilden, Germany) was used. DNA concentration and quality were assessed using the NanoDrop ND-1000 spectrophotometer (NanoDrop Technologies, Wilmington, DE, USA).

The V3-V4 hypervariable regions of the 16S rRNA gene were amplified using the 341F and 785R primes with Illumina adapter overhang sequences, as previously described [22]. PCR products were purified using a magnetic bead-based system (Agencourt AMPure XP; Beckman Coulter, Brea, CA, USA), followed by sample indexing through a limited-cycle PCR with Nextera technology. Indexed libraries were purified with another clean-up step, as described above, and then pooled at the equimolar concentration of 4 nM. Sequencing was performed by loading the denatured and 5 pM diluted library into the Illumina MiSeq platform, using the 2 × 250 bp paired-end protocol, per the manufacturer’s instructions (Illumina, San Diego, CA, USA).

### 2.3. Bioinformatic and Statistical Analysis

Raw sequences were processed using a pipeline combining PANDAseq [23] and QIIME 2 [24]. After length and quality filtering, sequences were binned into amplicon sequence variants (ASVs) using the DADA2 pipeline [25]. The taxonomic assignment was performed using the VSEARCH algorithm [26] against the Greengenes database (May 2013 release). Sequence reads were deposited in the National Center for Biotechnology Information Sequence Read Archive (NCBI SRA; BioProject ID PRJNA780429).

Alpha diversity was computed using several metrics, including the number of observed ASVs and the Shannon index. Beta diversity was estimated by weighted and unweighted UniFrac distances, which were used to build Principal Coordinates Analysis (PCoA) plots. All statistical analyses were performed on R and RStudio software. PCoA plots were constructed with the “Made4” [27] and “Vegan” (http://www.cran.r-project.org/package=vegan/packages, accessed on 26 February 2021) packages and data separation was tested by a permutation test with pseudo-F ratio (function “Adonis” in “Vegan”). To assess the significance of differences in GM alpha diversity and composition between groups, the Kruskal–Wallis test followed by post-hoc Wilcoxon tests (paired or unpaired as needed) was used. To reconstruct the trajectory over time (i.e., from the PRE to the P20 timepoint) of the major genera of GM in relation to the duration of FN, Circos plots were generated using the “Circos Table Viewer” software from Krzywinski et al. [28]. When appropriate, P values were corrected for multiple comparisons using the Benjamini–Hochberg method. A false discovery rate (FDR) ≤ 0.05 was considered as statistically significant.

Qualitative clinical variables were compared using Fisher’s exact or Pearson’s chi-square test, while means were compared using the Mann–Whitney U test. Cumulative incidences of BSI and GvHD were calculated using the Kalbfleisch and Prentice method and compared using the Gray test.

## 3. Results

### 3.1. Study Cohort Description

Thirty-seven patients were enrolled in the study. Clinical and transplant characteristics are summarized in Table 1.

Patients were stratified according to the median value of days of fever into two groups: less than or equal to three days (*n* = 16) and more than three days (*n* = 21). Clinical characteristics were comparable between the two groups (Table 2).

FN was treated according to local protocols, using broad-spectrum antibiotics as the first line. Details of infections for enrolled patients are reported in Appendix A. A detailed summary of the administered antibiotics is provided in Appendix A. No differences were found in the total number of days of antibiotic administration between patients with longer or shorter duration of fever (median days, 20.5 vs. 17.0, *p* = 0.1).

### 3.2. Gut Microbiota Dynamics in Pediatric Allo-HSCT Patients in Relation to the Duration of Febrile Neutropenia

A total of 161 fecal samples were collected prior to HSCT and at various timepoints after FN development, up to 30 days later, to reconstruct the GM trajectory (Figure 1).

Next-generation sequencing of the 16S rRNA gene yielded 25,563,052 high-quality reads (mean ± SD, 158,777 ± 52,330) clustered into 3279 ASVs. At baseline, the GM profile did not stratify for any of the possible confounding factors, i.e., transplant center, antibiotic prophylaxis, GvHD development, underlying disease, previous chemotherapy, and previous antibiotic therapies (permutation test with pseudo-F ratio, *p ≥* 0.08) (Appendix A). The GM dynamics were then reconstructed separately into the two groups of patients with different duration of fever (i.e., more vs. less than or equal to three days). According to both weighted and unweighted UniFrac metrics (Figure 2A; Appendix A), samples from patients with more than three days of fever deviated significantly from the baseline over time (*p* ≤ 0.05).

In contrast, samples from patients with shorter fever duration overlapped regardless of timepoint (*p* ≥ 0.4), suggesting some temporal stability of the GM structure from before transplant up to 30 days after the onset of FN. This opposite trend was also corroborated by the alpha diversity analysis (Figure 2B). In fact, both the Shannon index and the number of observed ASVs showed a significant loss of diversity at all post-FN timepoints compared to baseline, in patients with a longer fever duration (Wilcoxon test, *p* < 0.05). On the other hand, patients with less than or equal to three days of fever were characterized by decreased diversity only at P2 compared to PRE (*p* < 0.05).

From the taxonomic standpoint (Appendix A), as expected for pediatric allo-HSCT patients [7,15], the GM was overall dominated by the phylum Firmicutes (mean relative abundance across the dataset, 66.6%), and specifically by the *Enterococcaceae*, *Ruminococcaceae*, *Lachnospiraceae, Streptococcaceae*, *Lactobacillaceae* and *Erysipelotrichaceae* families. *Bifidobacteriaceae* and *Coriobacteriaceae,* both belonging to Actinobacteria (cumulative relative abundance, 12.6%), as well as *Enterobacteriaceae* and *Bacteroidaceae* (Proteobacteria and Bacteroidetes members, respectively; 11.8% and 5.9%) accounted for the remainder of the GM community. Consistent with the diversity data, when comparing the two groups of patients with different duration of fever, many compositional differences emerged (Figure 3).

In particular, the GM profile of patients with more than three days of fever showed a progressive thinning of subdominant genera, especially starting from TAKE, leaving room for the overabundance of potential pathobionts, such as *Enterococcus*. The proportions of the latter were in fact significantly higher at P20 than PRE in patients with more than three days of fever. On the other hand, the GM of patients with less than or equal to three days of fever was characterized by overall greater compositional stability over time, with only some early fluctuations in *Actinomyces* and *Staphylococcus*, which increased at P5 compared to PRE, and *Ruminococcus* whose relative abundance was lower at P5 and TAKE than PRE (*p* < 0.05). Finally, both patient groups showed a reduction over time in some health-associated, SCFA-producing GM members (i.e., *Coprococcus*, *Dorea*, *Ruminococcus*, *Roseburia*, *Blautia* and *Faecalibacterium*) (*p* < 0.05) (Appendix A).

### 3.3. Early and Late Gut Microbiota Signatures of Febrile Neutropenia Duration in Pediatric Allo-HSCT Patients

To identify potential signatures of fever duration, between-group comparisons were made at the different timepoints. According to our findings, *Collinsella*, *Megasphaera*, *Prevotella* and *Roseburia* were more represented in PRE samples from patients with more than three days of fever (Wilcoxon test, *p* < 0.05) (Figure 4A).

Patients with longer lasting fever were also distinguished by increased proportions of *Eggerthella* and *Akkermansia* at TAKE, as well as *Streptococcus* at P20 (*p* < 0.05) (Figure 4B) (Appendix A).

### 3.4. Association between Febrile Neutropenia Duration and Development of Other Transplant Complications

Finally, we investigated whether fever duration was associated with other clinical outcomes, namely BSI and GvHD. No significant differences were found between patients with longer or shorter fever duration for BSI onset (4/16 vs. 5/21, *p* = 0.9), as well as for any grade GvHD (10/16 vs. 7/21, *p* = 0.1) or grade II-IV aGvHD (4/16 vs. 4/21, *p* = 0.7) (Table 2). We then assessed the impact of FN duration on length of hospitalization and found that patients with longer fever duration had longer hospitalization (median in days: 61.3 vs. 40.0, *p* = 0.001).

## 4. Discussion

To our knowledge, this is the first report in the pediatric HSCT setting that has investigated the relationship between the GM dynamics and FN. According to our data, an overall more stable GM configuration over time, in terms of both alpha and beta diversity, is associated with a shorter duration of fever. Interestingly, this was achieved for three different centers across Europe, suggesting that this association is robust to the so-called geographic effect [29]. These findings are in line with previous reports, which suggest that a more diverse GM is associated with the reduction of transplant complications [6,30,31]. Consistently, patients with long-lasting fever presented greater compositional instability of GM, with in particular an overabundance of *Enterococcus*, frequently known as a possible pathobiont [32]. Moreover, we found that patients with fever of longer duration exhibited higher pre-HSCT levels of *Collinsella*, *Megasphaera*, *Prevotella* and *Roseburia*, which could therefore represent possible predictive signatures of FN duration. This latter finding is partially in contrast with what was reported in adult patients undergoing HSCT, in whom a higher amount of *Collinsella* and *Prevotella* was associated with the absence of fever during the neutropenic period [19]. However, it should be remembered that in the pediatric subject, the composition (and functionality) of the GM is very different, which probably leads to the establishment of different ecological networks that affect the host health in a different way. Furthermore, *Collinsella* is often reported in the literature as a pathobiont and associated with impaired integrity of the intestinal barrier [33], which may support its role in the development of fever. The GM profile of patients with longer fever duration was also enriched in other potentially harmful components, such as *Akkermansia*, *Eggerthella* and *Streptococcus*. The former is a mucus degrader, previously associated with other pathological conditions, such as inflammatory bowel disease [34], colonic epithelial carcinogenesis [35] and multiple sclerosis [36]. Notably, the presence of *Akkermansia* has been related with the onset of FN in adults undergoing intensive chemotherapy, and suggested to contribute to oxidative stress as a possible mediator of FN [20]. On the other hand, *Akkermansia* has also been shown to have beneficial effects in other clinical settings, such as in patients with obesity [37] or in the response rate of cancer mice treated with anti-PD1 drugs [38]. Thus, its role should be better addressed in future studies.

While our data are insufficient to postulate any causal relationship between GM and FN duration in pediatric HSCT recipients, we can hypothesize that GM dysbiosis may be associated with a prolonged inflammatory state. An altered intestinal ecosystem, enriched in pathobionts and mucus degraders with an impoverished commensal community, can worsen the intestinal mucosa damage, allowing the translocation of microbial molecules and microbes [39], thus sustaining a systemic inflammatory process and leading to a longer duration of fever. Indirect markers to assess mucosal and endothelial damage, such as citrulline, lipopolysaccharides (LPS), regenerating islet-derived protein 3 alpha (REG-3-alpha) or urinary 3-indoxil-sulfate (3-IS), should be analyzed in the future alongside GM trajectories to further address this hypothesis [40].

On the other hand, the clinical impact of FN duration remains to be fully assessed. In fact, it must also be said that whether fever per se can be considered as a complication is still debated. In our cohort, we were unable to find a relationship between fever and BSI or GvHD, but we did find a difference in the length of hospitalization. Moreover, no differences in antibiotic exposure were found between the two groups. Probably, the number of enrolled patients was too small to draw firm conclusions, and this represents a limitation of this study. Another limitation is the enrolment of patients undergoing HSCT for any indications, thus representing a possible confounding factor. Lastly, assessing FN may be subject to a certain degree of bias, as it is known to be due to different causes. Therefore, confirmation from larger cohorts is highly awaited.

It has been reported that patients with FN experience non-specific symptoms that may impair general conditions and prolong hospitalization [41], as in our case, and thus targeting FN may present clinical benefits. While our study also included patients with non-oncological diseases, improving HSCT certainly has a clinical impact on cancer patients, representing a key treatment for high-risk patients. In fact, alongside the new drugs in development [42], HSCT represents the best therapeutic option also for cancer patients not responsive to conventional therapies.

## 5. Conclusions

In summary, we provided the first evidence of a correlation between FN duration and GM dynamics in pediatric patients undergoing HSCT. Future studies are needed to better define the biological mechanisms underpinning this relationship. Besides, studies on larger cohorts are needed to assess the clinical impact of FN duration, and whether novel approaches targeting the GM ecosystem [12,43] could be clinically relevant.

## Figures and Tables

**Figure 1 cancers-14-01932-f001:**
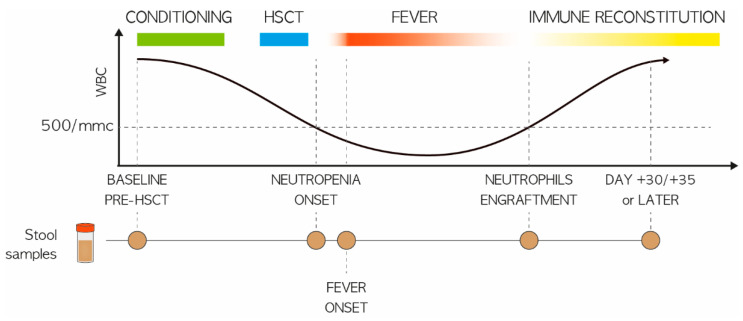
Study design. Schematic representation of fecal sampling for pediatric patients undergoing HSCT, in relation to the development of FN. Circles indicate the sampling timepoints, i.e., at baseline (before transplant; PRE), at the onset of neutropenia (day −2/+2; P2), at the onset of fever (day +4/+5; P5), at engraftment (TAKE) and after engraftment (day +20/+30; P20). Patients were stratified based on the median of total fever days into two groups: (i) less than or equal to three days (*n* = 16) and (ii) more than three days (*n* = 21).

**Figure 2 cancers-14-01932-f002:**
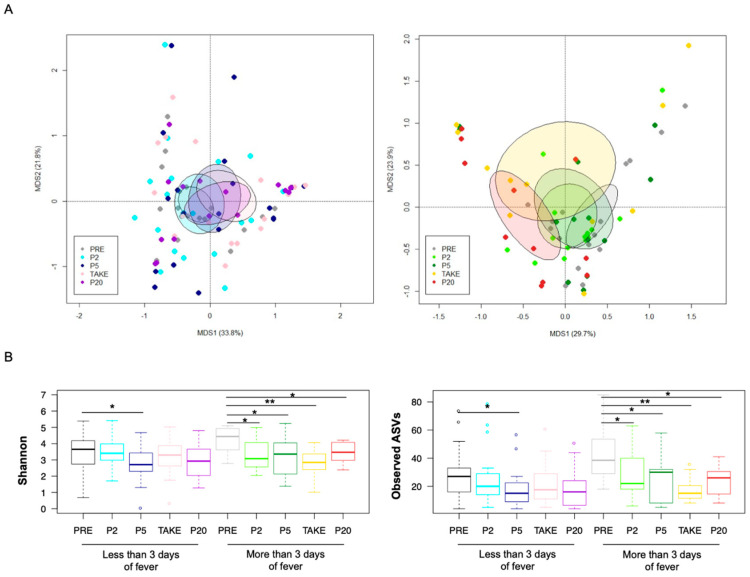
Gut microbiota diversity in pediatric allo-HSCT patients with different duration of febrile neutropenia. (**A**) PCoA based on weighted UniFrac distances between the gut microbiota profiles of samples collected before transplant (PRE), at the onset of neutropenia (day −2/+2; P2), at the onset of fever (day +4/+5; P5), at engraftment (TAKE) and after engraftment (day +20/+30; P20), in pediatric allo-HSCT patients with less than or equal to (**left**) and more (**right**) than three days of fever. Ellipses include 95% confidence area based on the standard error of the weighted average of sample coordinates. Significant separation among groups was found only for patients with longer fever duration (permutation test with pseudo-F ratio, *p* = 0.03). See also Appendix A; (**B**) Boxplots showing the dynamics of alpha diversity, estimated with the Shannon index (**left**) and the number of observed ASVs (**right**), in patients with less than or equal to vs. more than three days of fever. Wilcoxon test, * for *p* < 0.05, ** for *p* < 0.01.

**Figure 3 cancers-14-01932-f003:**
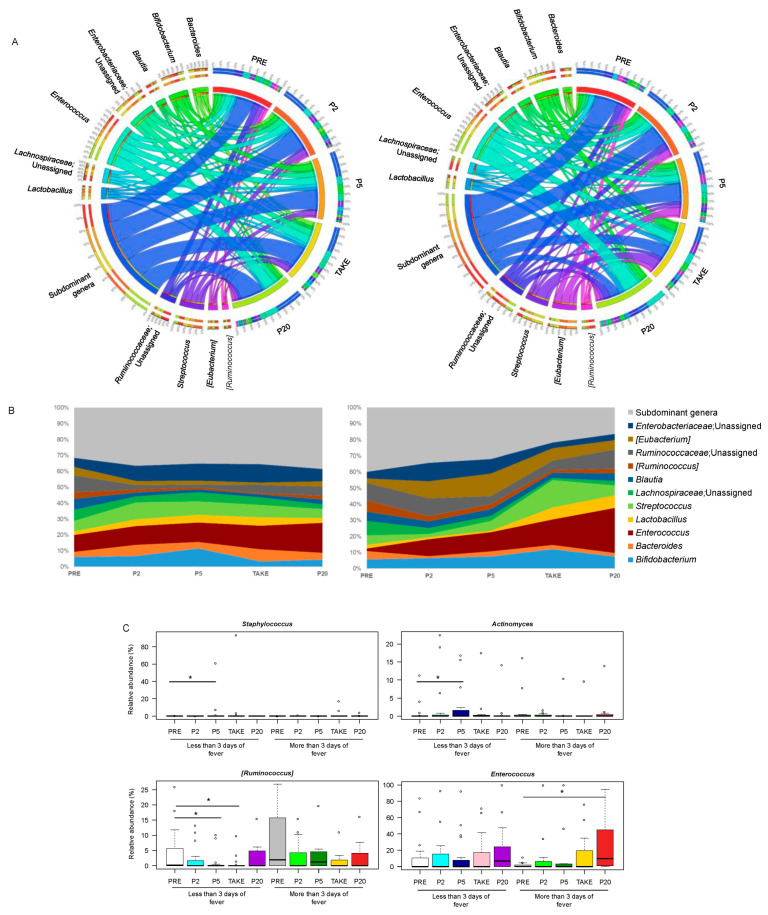
Genus-level gut microbiota trajectory in pediatric allo-HSCT patients with different duration of febrile neutropenia. Circos (**A**) and area (**B**) plots showing the relative abundance over time of the major genera in the GM of patients with less than or equal to (**left**) vs. more (**right**) than three days of fever. Only taxa with mean relative abundance > 20% in at least five samples of the total dataset are shown. PRE, before transplant; P2, at the onset of neutropenia (day −2/+2); P5, at the onset of fever (day +4/+5); TAKE, at engraftment; P20, after engraftment (day +20/+30); (**C**) Boxplots showing the relative abundance distribution of genera significantly differentially represented over time in patients with different duration of fever. Wilcoxon test, * for *p* < 0.05.

**Figure 4 cancers-14-01932-f004:**
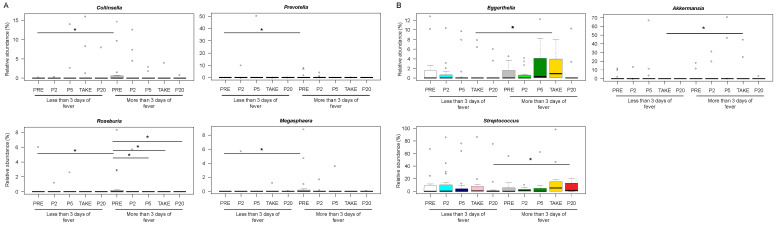
Early and late gut microbiota signatures of febrile neutropenia duration. Boxplots showing the relative abundance distribution of genera significantly differentially represented between patients with less than or equal to vs. more than three days of fever, before transplant (**A**) and after the onset of fever (**B**). PRE, before transplant; P2, at the onset of neutropenia (day −2/+2); P5, at the onset of fever (day +4/+5); TAKE, at engraftment; P20, after engraftment (day +20/+30). Wilcoxon test, * for *p* < 0.05.

**Table 1 cancers-14-01932-t001:** Clinical and transplant characteristics of enrolled patients. Data are shown for the whole cohort and for each of the three pediatric centers. BM: Bone marrow; MAC: Myeloablative conditioning; MMUD: Mismatched unrelated donor; MSD: Matched sibling donor; MUD: Matched unrelated donor; PBSC: Peripheral blood stem cells; RIC: Reduced intensity conditioning.

Characteristic	Overall (*N* = 37)	Bologna (*N* = 12)	Verona (*N* = 12)	Wroclaw (*N* = 13)
Age at HSCT—year	8.3 (1.0–18.0)	8.8 (1.1–18.0)	9.0 (1.2–17.6)	7.2 (1.0–13.7)
Malignant disease—no. (%)	22 (59)	8 (67)	8 (67)	6 (46)
Donor—no. (%)				
MUD	20 (54)	6 (50)	7 (58)	7 (54)
MMUD	7 (19)	2 (17)	4 (33)	1 (8)
Haplo	5 (14)	3 (25)	0 (19)	2 (15)
MSD	5 (14)	1 (8)	1 (8)	3 (23)
Graft type—no. (%)				
BM	22 (60)	9 (75)	8 (67)	5 (38)
PBSC	15 (40)	3 (25)	4 (33)	8 (62)
Intensity of conditioning regimen—no. (%)				
MAC	24 (65)	10 (83)	7 (58)	7 (54)
RIC	13 (35)	2 (17)	5 (42)	6 (46)

**Table 2 cancers-14-01932-t002:** Comparison of clinical confounders between patients with longer and shorter duration of neutropenic fever. BM: Bone marrow; EN: Enteral nutrition; MAC: Myeloablative conditioning; MMUD: Mismatched unrelated donor; MSD: Matched sibling donor; MUD: Matched unrelated donor; PBSC: Peripheral blood stem cells; PN: Parenteral nutrition; RIC: Reduced intensity conditioning. Qualitative clinical variables were compared using Fisher’s exact or Pearson’s chi-square test.

	Fever > 3 Days (*n* = 16)	Fever ≤ 3 Days (*n* = 21)	*p*
Center:			0.136
Bologna	8 (50.0%)	4 (19.0%)	
Wroclaw	4 (25.0%)	9 (42.9%)	
Verona	4 (25.0%)	8 (38.1%)	
Underlying disease:			0.742
Malignant	10 (62.5%)	12 (57.1%)	
Non-malignant	6 (37.5%)	9 (42.9%)	
Antibiotic prophylaxis:			0.141
Yes	6 (37.5%)	13 (61.9%)	
No	10 (62.5%)	8 (38.1%)	
Type of conditioning:			0.260
MAC	12 (75.0%)	12 (57.1%)	
RIC	4 (25.0%)	9 (42.9%)	
Use of granulocyte colony-stimulating factor (G-CSF):			0.384
Yes	14 (87.5%)	16 (76.2%)	
No	2 (12.5%)	5 (23.8%)	
Corticosteroid for GvHD:			0.208
Yes	8 (50.0%)	7 (33.3%)	
No	2 (12.5%)	0 (0%)	
Bloodstream infections:			0.384
Yes	2 (12.5%)	5 (23.8%)	
No	14 (87.5%)	16 (76.2%)	
GvHD (any grade):			0.104
Yes	10 (62.5%)	7 (33.3%)	
No	6 (37.5%)	14 (66.7%)	
GvHD (grade II-IV):			0.705
Yes	4 (25.0%)	4 (19.0%)	
No	12 (75.0%)	17 (81.0%)	
Nutrition:			1.00
EN	6 (37.5%)	8 (38.1%)	
PN	10 (62.5%%)	13 (61.9%)	
Mucositis:			0.733
Grade 3–4	7 (43.8%)	7 (33.3%)	
Grade ≤ 2	9 (56.2%)	14 (66.7%)	
Graft source:			0.176
BM	12 (75.0%)	10 (47.6%)	
PBSC	4 (25.0%)	11 (52.4%)	
Donor:			0.460
MUD	7 (43.8%)	13 (61.9%)	
MMUD	4 (25.0%)	4 (19.0%)	
Haplo	4 (25.0%)	2 (9.5%)	
MSD	1 (6.2%)	3 (14.3%)	

## Data Availability

Sequence reads were deposited in the National Center for Biotechnology Information Sequence Read Archive (NCBI SRA; BioProject ID PRJNA780429).

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
