# Peer review of "Febrile Neutropenia Duration Is Associated with the Severity of Gut Microbiota Dysbiosis in Pediatric Allogeneic Hematopoietic Stem Cell Transplantation Recipients"

_cancers, 2022, doi:10.3390/cancers14081932_

Round 1

Reviewer 1 Report

The authors report a prospective, multi-center study in  pediatric patients undergoing allo HSCT where the relationship between the Gut microbiota (GM) dynamics and febrile neutropenia (FN) were investigated. they concluded that a more stable GM configuration over time is associated with a shorter duration of fever. The manuscript is well written.

Yet, I have some minor comments:

1. Patients could have received allo-HSCT for any diagnosis, with any conditioning, and with any donor source with some recieving prophylactic antibiotics and others not. This makes the interpretation of data more difficult as so many variables - besides all HSCT- might impact GM. This limitation should be mentioned in the discussion.

2. As expected, FN was almost universally present, no differences were found in the total number of days of antibiotic administration between patients with longer or shorter duration of fever, and FN had no impact on outcome. These observations raise the question of whether we really need predictive signatures of FN duration in terms of GM? I would very much like the authors to introduce in the discussion the significance of their findings and put them into context with the clincal outcomes. 

In light of the above mentioned, I would suggest to the authors to critically question their own conclusion and rephrase the following sentence: "These results 
could also suggest that the development of novel approaches targeting the GM ecosystem, such as nutritional modulation or the administration of live protective microorganisms or 
their metabolites[12,46], could potentially prevent or mitigate the onset of FN during the HSCT setting." 

3. Apart from GM, what were the other causes of FM identified in the patients besides blood stream infections including the location of infection (long, catheter, UTI..etc) and the pathogenic organisms identified (microbia, viral, and/ or fungal) with their corresponding numbers (preferably in a table)

4. Table 2: please add % in brackets.

Author Response

Reviewer #1:

The authors report a prospective, multi-center study in pediatric patients undergoing allo HSCT where the relationship between the Gut microbiota (GM) dynamics and febrile neutropenia (FN) were investigated. they concluded that a more stable GM configuration over time is associated with a shorter duration of fever. The manuscript is well written.

We would like to thank Reviewer #1 for appreciating our work and very valuable comments.

Yet, I have some minor comments:

  1. Patients could have received allo-HSCT for any diagnosis, with any conditioning, and with any donor source with some recieving prophylactic antibiotics and others not. This makes the interpretation of data more difficult as so many variables - besides all HSCT- might impact GM. This limitation should be mentioned in the discussion.

We agree with Reviewer #1 and we thank her/him for this crucial consideration. We would like to underline that this relative heterogeneity is quite mitigated by the homogeneity between the two study groups. We have added this specification in the discussion.

In any case, as suggested by Reviewer #2, we evaluated the impact of other factors on the GM structure at baseline, such as underlying disease, previous chemotherapy, and previous antibiotic therapies. Interestingly, no significant separation was found for any of these variables (please, see the new Supplementary Figure 1).

  1. As expected, FN was almost universally present, no differences were found in the total number of days of antibiotic administration between patients with longer or shorter duration of fever, and FN had no impact on outcome. These observations raise the question of whether we really need predictive signatures of FN duration in terms of GM? I would very much like the authors to introduce in the discussion the significance of their findings and put them into context with the clincal outcomes.

We agree with the raised issue, and we believe this to be a critical consideration. Indeed, our findings fail to correlate FN data with clinical outcomes (apart from duration of hospitalization) and this may be due to the relatively small cohort, which also includes patients from three distinct centers. Certainly, confirmation on a larger cohort is needed to conclusively assess the clinical implications of a predictive signature of FN. However, our results support a relationship between fever and GM profile, which represents another novel step forward in understanding this complicate crosstalk. In line with this valuable suggestion, we have modified the discussion.

In light of the above mentioned, I would suggest to the authors to critically question their own conclusion and rephrase the following sentence: "These results

could also suggest that the development of novel approaches targeting the GM ecosystem, such as nutritional modulation or the administration of live protective microorganisms or

their metabolites[12,46], could potentially prevent or mitigate the onset of FN during the HSCT setting."

We have modified the sentence as suggested by Reviewer #1.

  1. Apart from GM, what were the other causes of FM identified in the patients besides blood stream infections including the location of infection (long, catheter, UTI..etc) and the pathogenic organisms identified (microbia, viral, and/ or fungal) with their corresponding numbers (preferably in a table)

We would like to thank Reviewer #1 for this constructive suggestion. We have added a new Supplementary Table 1 listing the proven bacterial, fungal and viral infections in the two groups. Moreover, we have categorized the pathogens according to the Common Commensal tab of the NHSN Organisms List.

  1. Table 2: please add % in brackets.

We have added % in brackets in Table 2.

Reviewer 2 Report

The Authors report here a longitudinal study assessing the relationship between gut microbiote composition and duration of febrile neutropenia in a cohort of pediatric patients receiving allogeneic stem cell transplantation. In recent years, the role of gut microbiome in complications after SCT has become more and more predominant both in the adult and in the pediatric setting.
While the association between gut microbiome composition and febrile neutropenia has  been explored deeper in other settings, data in the HSCT pediatric population are lacking. In this context, this study represents a first experience reporting results globally consistent with evidence published in the adult setting.
The article is well organized and table and graphics are clear and easy understandable. In particular, the analysis of the microbiome diversity is well performed and informative. Results sound interesting, although a higher number of patients is required to draw definitive conclusions.
In my opinion, there are few points to be clarifyed:
1) Using febrile neutropenia duration as primary endopoint is subject to a certain degree of bias, as febrile neutropenia early after transplant is known to be due to many different causes and influenced by many variables. It should be better specified if other specific infections (i.e. fungal, viral) or causes of fever other BSI were found, and the analysis should hold true especially for BSI and febrile neutropenia without microbiologic identifications (possibly reflecting intestinal translocations).
2) also, more importance should be stressed on patients outcome, in particular if longer fever was associated with a prolongation of hospitalisation or a worse outcome in this cohort of patients
3) Did you study other factors than transplant center, antibiotic prophylaxis and GvHD development for diversity in GM composition at baseline? I.e, underlying disease, previous chemotherapy cycles or previous infections/repeated antibiotic therapies are all variables that can impact on outcome and febrile neutropenia duration.
4) Could you specify bacterias causing BSI? Were they possibily translocated from gastroenteric tract?

Author Response

Reviewer #2:

The Authors report here a longitudinal study assessing the relationship between gut microbiote composition and duration of febrile neutropenia in a cohort of pediatric patients receiving allogeneic stem cell transplantation. In recent years, the role of gut microbiome in complications after SCT has become more and more predominant both in the adult and in the pediatric setting.

While the association between gut microbiome composition and febrile neutropenia has been explored deeper in other settings, data in the HSCT pediatric population are lacking. In this context, this study represents a first experience reporting results globally consistent with evidence published in the adult setting.

The article is well organized and table and graphics are clear and easy understandable. In particular, the analysis of the microbiome diversity is well performed and informative. Results sound interesting, although a higher number of patients is required to draw definitive conclusions.

We would like to thank Reviewer #2 for the words of appreciation. We totally agree with the consideration that a higher number of patients should be preferable, and this is the goal of future studies. Unfortunately, reaching such numbers even in multicenter studies in children is extremely challenging. In our opinion, while our study does not provide a firm conclusion, it still provides important data on an interesting relationship that should be further addressed.

In my opinion, there are few points to be clarifyed:

1) Using febrile neutropenia duration as primary endopoint is subject to a certain degree of bias, as febrile neutropenia early after transplant is known to be due to many different causes and influenced by many variables. It should be better specified if other specific infections (i.e. fungal, viral) or causes of fever other BSI were found, and the analysis should hold true especially for BSI and febrile neutropenia without microbiologic identifications (possibly reflecting intestinal translocations).

We agree with Reviewer #2 that febrile neutropenia is subject to a certain degree of bias and this represents a limitation of the study. We have added this specification in the discussion. In any case, as also suggested by Reviewer #1, we have provided a more detailed characterization of the infections in the two groups, in terms of bacterial, fungal and viral pathogens that were found (please, see the new Supplementary Table 1).

2) also, more importance should be stressed on patients outcome, in particular if longer fever was associated with a prolongation of hospitalisation or a worse outcome in this cohort of patients

We would like to thank Reviewer #2 for this valuable suggestion. We compared the days of hospitalization between the two groups and found that patients experiencing a shorter duration of FN were hospitalized for a shorter period (p=.001). Unfortunately, regarding the other outcomes, the small number of patients and the relatively short follow-up did not allow us to perform the analysis. For sure, we will specifically address this point in future studies. We have modified the text accordingly.

3) Did you study other factors than transplant center, antibiotic prophylaxis and GvHD development for diversity in GM composition at baseline? I.e, underlying disease, previous chemotherapy cycles or previous infections/repeated antibiotic therapies are all variables that can impact on outcome and febrile neutropenia duration.

As suggested by the Reviewer, we further explored the GM structure at baseline based on the underlying disease, previous chemotherapy and antibiotics used in the previous 45 days (we chose this cut-off based on the publication by Palleja et al., 2018, Nature Microbiology). According to a PCoA analysis based on UniFrac distances, the baseline GM profiles did not segregate for any of the possible confounding variables. We have added these data to Supplementary Figure 1 and modified the figure legend accordingly. 

4) Could you specify bacterias causing BSI? Were they possibily translocated from gastroenteric tract?

We have added this information in the new Supplementary Table 1. Based on the Common Commensal tab of the NHSN Organisms List, the bacteria identified appeared to be mainly common commensals and organisms categorized for urinary tract infection. Only S. oralis and P. mirabilis may result from translocation from mucosal damage.

Reviewer 3 Report

The article "Febrile neutropenia duration is associated with the severity of gut microbiota dysbiosis in pediatric allogeneic hematopoietic stem cell transplantation recipients" is well-written and well-organized but I think that it does not fit the scope of the journal "Cancers" since not all patients have malignant diseases and the focus of the paper is not cancer.

I suggest the authors to submit it to another journal which fits the scope.

Author Response

Reviewer #3: 
The article "Febrile neutropenia duration is associated with the severity of gut microbiota dysbiosis in pediatric allogeneic hematopoietic stem cell transplantation recipients" is well-written and well-organized

We would like to thank Reviewer #3 for these appreciative words.

I think that it does not fit the scope of the journal "Cancers" since not all patients have malignant diseases and the focus of the paper is not cancer. I suggest the authors to submit it to another journal which fits the scope.

We do agree with Reviewer #3 that not all patients included in the study were affected by malignant diseases. However, considering that hematopoietic stem cell transplantation is a therapeutic strategy performed for most high-risk malignant diseases, we believe that the content of our work is still within the journal scope. Moreover, more than half of the patients in our study were affected by malignant diseases. Lastly, Cancers journal has already published papers on hematopoietic stem cell transplantation in children with both malignant and non-malignant diseases (https://www.mdpi.com/2072-6694/13/12/3090, DOI 10.3390/cancers13123090).

Round 2

Reviewer 3 Report

Thank you for your answers to my comments on the relevance of the paper for cancer treatment.

Author Response

We would like to thank Reviewer #3 for his reply. Following his suggestion and Academic Editor’s one, we added in the Introduction (line 84-86) and in the Discussion (line 357-359) a brief specification on the pertinence of the results for Journals Scopes. In particular, we specified that improving HSCT, both in oncological and non-oncological patients, can have a strong clinical impact on the formers.
